

# Prediction model of stock return on investment based on hybrid DNN and TabNet model

Tonghui Zhang[1], Ming Da Huo[2], Zhaozhao Ma[3], Jiajun Hu[4], Qian Liang[5] and Heng Chen[6]

[1] Brooks School of Public Policy, Cornell University, Ithaca, NY, United States of America
[2] Jinan University, Guangzhou, Guangdong, China
[3] Central South University, Changsha, Hunan, China
[4] National University of Singapore, Singapore, Singapore, Singapore
[5] Yunnan Normal University, Kunming, Yunnan, China
[6] Fuzhou software vocational and technical college, Fuzhou, Fujian, China

Corresponding author
Heng Chen, hc635@nau.edu

## ABSTRACT

With the development of the social economy, research on stock market prediction is in full swing. However, the fluctuations in stock price and returns are influenced by many factors, including political policies, market environment, investor psychology, and so on. The traditional analysis method, based on subjective experience, requires significant time and effort, and its prediction accuracy is often poor. Now, the application of machine learning algorithms to predict stock returns has become a hot topic among scholars. This article comprehensively analyzes the advantages and disadvantages of support vector machine (SVM), tree-based algorithms, and neural network algorithms in processing tabular data and time series data. It proposes a hybrid model based on the deep neural network (DNN) and TabNet models, combining the strengths of the DNN and tree-based models. In the model training stage, two neural networks are established to accept the inputs of ID features and numerical features, respectively, and multiple fully connected layers are used to complete the construction of the DNN model. The TabNet is implemented based on the attention transformer and feature transformer, and the prediction results of the two models are fused. The proposed model has a best Pearson correlation coefficient (PCC) value and a lowest root mean square error (RMSE) value at the same time, because the hybrid algorithm performs particularly well on large data sets with the least feature engineering and has strong interpretability, such as quantifying the contribution of different features in the model, it has certain theoretical significance and wide application value.

## INTRODUCTION

Nowadays, the financial market is replete with a growing array of financial products and derivatives, including bonds, futures, funds, stocks, and so on. Stocks, in particular, epitomize high-risk and high-return investments, reflecting to some extent the operation

of the national economy and influencing the direction of many economic activities. On the one hand, stocks serve as instruments for capital financing and investment, enabling listed companies to raise funds conveniently in both domestic and international markets. On the other hand, an increasing number of investors are flocking to the stock market. To invest more effectively, many investment institutions engage in stock investment research through expert analysis, portfolio methods, strategy research, and other approaches that rely on subjective experience. However, the stock market environment is exceedingly complex, with stock price trends and returns influenced by numerous factors, including political policies, the international environment, enterprise operations, and so on. For instance, the COVID-19 pandemic in 2020 had a profound impact on the financial market, further exacerbating the complexity and uncertainty of stock market risks. Traditional stock analysis methods, such as the Elliott wave principle and Dow theory, can assess future stock fluctuations to a certain extent, but they rely too much on subjective judgment, compromising their reliability. Only by uncovering the hidden laws of market fluctuation through extensive data analysis and accurately predicting stock returns can investors hope to achieve higher returns from the stock market.

Machine learning algorithm for stock prediction: Due to the continuous development and iteration of computer performance and machine learning technology, artificial intelligence methods may come to the field of financial analysis with massive data. In recent years, increasing number of machine learning algorithms for predicting the stock return and price are developed by researchers from both academia and industry. The support vector machine (SVM) algorithm is a research hotspot with a mature theory and good generalization ability. It has been widely used in financial trend analysis and projections, such as predicting stock prices and returns (*Huang, Wei & Zhou, 2022*; *Ali et al., 2021*). The SVM algorithm is used to predict future trends, trading volumes, profit margins, and other indicators of the stock market (*Doroudyan & Niaki, 2021*), with prediction accuracy that is ahead of traditional time series methods (*Tas & Atli, 2022*). *Dai & Ning (2011)* constructed a model based on the SVM algorithm to predict stock prices using relevant financial news and achieved good results. *Devi, Bhaskaran & Kumar (2015)* comprehensively considered the impact of different SVM parameters on the prediction results of the model, and combined this with Cuckoo Search (CS) (*Zhi et al., 2021*) technology to adjust and optimize the model parameters. The results show that the hybrid CS-SVM model performs better than the traditional SVM model in the task of stock price prediction (*Devi, Bhaskaran & Kumar, 2015*). Moreover, the SVM algorithm with different kernel functions and parameters greatly impacts the accuracy of prediction results (*Chhajer, Shah & Kshirsagar, 2022*). *Zhang (2007)* studied and predicted stock fluctuations comprehensively and meticulously using the SVM algorithm. The results indicate that predicting a single stock involves great uncertainty, but the prediction accuracy for the index can reach 60%. However, when the sample data is increased, optimizing the solution becomes difficult and the model unstable (*Zhang, 2007*). In summary, although SVM is adept at tackling the nonlinear binary classification problem with small samples, there are still some deficiencies in the massive data stock prediction problem. Additionally, the SVM algorithm is similar to a black box model and cannot explain the reasons for the

selected parameters. Therefore, there is significant room for improvement in the research and forecasting of stock returns.

With the application of data mining technology in the field of financial intelligence, the decision tree has consistently played an increasingly important role (*Bansal, Goyal & Choudhary, 2022*). The algorithm is based on the principle of minimizing the loss function and can be regarded as a top-down recursive process. This algorithm and its derived related algorithms, including Random Forest (RF), and gradient boosting decision tree (GBDT) are considered to be the best and most commonly used supervised learning algorithms. These algorithms possess strong explanatory and self-learning abilities and have been widely used to solve various data science problems (*Luo et al., 2022*). *Awan et al (2021)* studied the stocks of the top 10 companies, which included historical stock prices, using models such as linear regression and RF. *Karim, Alam & Hossain (2021)* proposes a system that operates using two approaches: linear regression and decision tree regression. These two models are employed to analyze datasets of various sizes to assess the accuracy of stock price prediction forecasts. In *Sadorsky (2021)*, the machine learning technique known as RF is used to predict the direction of stock prices for clean energy exchange-traded funds.

The XGBoost algorithm is an important component within ensemble learning algorithms and is an efficient implementation of GBDT. It offers higher prediction accuracy and operational efficiency, and its robustness has also been improved to a certain extent (*Chen & Guestrin, 2016*). *Wang & Guo (2019)* used the Grid Search algorithm to optimize the parameters of XGBoost and established a stock price prediction model. Compared with the GBDT and SVM models, they obtained more accurate prediction results (*Wang & Guo, 2019*). *Yun, Yoon & Won (2021)* studied the influence of key indicators and feature engineering on the prediction results of the XGBoost algorithm. Experiments verified the importance of feature engineering in stock price prediction models. For instance, feature expansion could significantly improve the prediction accuracy of the XGBoost algorithm (*Chen et al., 2018*). The light gradient boosting machine (LightGBM) algorithm can be regarded as an enhanced version of the XGBoost algorithm, which uses a histogram-based algorithm to replace the original greedy algorithm. While ensuring that accuracy does not decrease, the LightGBM algorithm greatly reduces memory usage and evidently improves operational efficiency (*Ke et al., 2017*). *Qu, Zhang & Qin (2020)* selected the LightGBM algorithm to predict the trend of stock price changes over 10 days and conducted denoising and detailed feature engineering processing on the data before the experiment. The results indicate that the algorithm can effectively predict the short-term development trend of stock prices (*Qu, Zhang & Qin, 2020*). *Xiaosong & Qiangfu (2021)* established a prediction model for stock prices and returns based on the LightGBM algorithm and enhanced the accuracy of the results by selecting appropriate features. The results demonstrate that this algorithm outperforms both the XGBoost and RF algorithms in predicting stock prices (*Xiaosong & Qiangfu, 2021*). These tree-based models exhibit good accuracy and high efficiency in the task of predicting stock prices or returns, but at the same time, their high performance largely depends on the quality of data and feature engineering.

Deep learning algorithm for stock prediction: Currently, there are several algorithms developed within the neural network framework that exhibit strong nonlinear generalization capabilities and adaptability. These algorithms include the back propagation neural network (BPNN), the wavelet neural network (WNN), and the deep neural network (DNN), among others. They have provided new ideas and methods for predicting stock prices and returns and have gained wide application in the field of financial intelligence (*Kimoto et al., 2020*).

The BPNN is a widely-used neural network structure. It has been utilized to create models for predicting stock prices (*Yixin & Zhang, 2020*; *Huang, Wu & Li, 2021*), and the results indicate that prediction accuracy is higher when stock price fluctuations are minimal. However, with increased volatility, the predictive outcomes can significantly deviate from the actual prices. To overcome the limitations of BPNN, such as slow convergence speed and its tendency to get trapped in local minima, researchers have applied various optimization algorithms to enhance and refine the model.

For example, the use of genetic algorithms to optimize and adjust the initial weight thresholds of the BPNN has been found to effectively enhance its prediction accuracy and generalization ability (*Ding, Su & Yu, 2021*; *Liu, Han & Wang, 2023*). *Liu & Hou (2021)* utilized the Bayesian Regularization (BR) algorithm to optimize the fitness function of BPNN. Experimental results in stock prediction demonstrated the effectiveness of this approach; specifically, the prediction accuracy of BR-BPNN was found to be 42.81% higher than that of the ordinary BPNN (*Liu & Hou, 2021*).

Wavelet analysis, with its powerful multi-scale resolution capability, has been combined with neural networks to identify different frequencies in stock data sequences and to capture their changing trends (*Ye & Wei, 2020*). Some researchers have developed stock market prediction models based on WNN and have optimized the initial parameters and related settings of the models using techniques such as CS (*Zhi et al., 2021*), particle swarm optimization (*Gang et al., 2021*), and genetic algorithms (*Fang et al., 2022*), leading to improved prediction accuracy.

With the emergence of deep learning theory and advancements in computing technology, algorithms such as DNN, convolutional neural networks (CNN), and recurrent neural networks (RNN) have experienced rapid progress. These algorithms have been widely employed by researchers to develop more complex and realistic models for predicting financial data trends (*Jiang, 2021*).

Deep learning methods have the inherent capability to automatically extract high-level features from raw data, eliminating the need for the laborious feature engineering processes (*Wang & Fan, 2021*). Furthermore, these methods can handle and process large volumes of data effectively, leading to more accurate predictions (*Akita et al., 2021*).

For example, in the field of stock price prediction, researchers have applied deep learning techniques to construct advanced models. *Bhardwaj (2021)* implemented a CNN model to forecast stock price fluctuations. The results indicated that the model achieved higher accuracy in predicting upward trends compared to downward trends. *Mahajan (2022)* proposed a hybrid approach by combining quantum neural networks (QNN) with CNN

to address the resource-intensive nature of CNNs. The combined model demonstrated promising results in stock price prediction.

In another study, *Chen & Huang (2021)* developed an optimized CNN model to predict future trends in the stock market using various input features. The model achieved an accuracy of 67% in predicting stock price movements over the next ten days, resulting in a return on investment of 13.23%, which surpassed the performance of other algorithms.

In summary, advancements in deep learning algorithms, such as DNN, CNN, and RNN, have facilitated the development of more sophisticated models for forecasting financial data. These models provide benefits such as automated feature extraction, scalability to large datasets, and enhanced prediction accuracy.

Data related to stock prices or returns can be regarded as a typical time series. The outputs of BPNN, CNN, and other models only consider the influence of the previous input and ignore the influence of inputs at other times, resulting in the loss of information, especially in time series. The RNN algorithm can extract the time sequence in the data and store the information from previous times for the analysis and calculation of the current output (*Vargas et al., 2021*). Compared with the CNN algorithm, RNN-based models have better applications in time series analysis and prediction. The long short-term memory (LSTM) algorithm is currently the most used time series algorithm based on RNN, which overcomes the problems of gradient disappearance or explosion found in traditional RNN algorithms. This algorithm is a time-recursive neural network, which is suitable for processing and predicting important events with long intervals and delays in time series, resulting in a high degree of consistency with stock prediction (*Bao et al., 2020*). *Kumar & Ningombam (2018)* predicted stock prices based on the LSTM model and verified the applicability and effectiveness of this algorithm. *Shin, Choi & Kim (2017)* predicted stock price trends based on different deep learning models. The results showed that the LSTM model had a faster learning speed, more stable performance, and the prediction accuracy was about 15% higher than that of the original DNN model.

However, compared with tree-based models, such as XGBoost and LightGBM, the algorithms based on DNN are not satisfactory in the processing and prediction of tabular data (*Huang & Xie, 2018*), even with sufficient feature engineering. The TabNet algorithm uses the DNN structure to construct the decision manifold of the tree model, which not only retains the end-to-end and representation learning characteristics of DNNs but also has the advantages of tree model interpretability and sparse feature selection (*Arik & Pfister, 2021*). This algorithm can achieve the same or even higher performance as the mainstream tree-based models in the task of tabular data prediction and can reduce or eliminate the impact of feature engineering on the results (*Borghini & Giannetti, 2021*; *Shah, Du & Xu, 2022*), becoming a research hotspot in the current study of tabular data processing (*Yan et al., 2021*). *Sun & Yang (2022)* used the TabNet model to predict the trend of the China Securities Index (CSI) 300, and the results showed that the model improved upon the low accuracy issues of tree-based models and the tendency of DNNs to overfit. The performance of this model was better than that of LSTM, gate recurrent unit (GRU), and other models. Additionally, this model has better interpretability than DNNs. *Zheng (2022)* proposed a method to predict the price trend of futures contracts based on the TabNet model, using

technical indicators and customer information as input data. Ultimately, a prediction accuracy of 60.1% was obtained, verifying the reliability of the algorithm.

In the prediction of stock price or return, regardless of which algorithm or model is used, there will be certain predictive limitations, and the data and information utilized are also limited. Therefore, in this article, we integrate the TabNet algorithm with the DNN algorithm to build a new stock return prediction model to achieve better performance. The prediction results are compared with those from XGBoost, LightGBM, and other algorithms. The results show that our model performs better in terms of interpretability, stability, and prediction accuracy.

Main contributions: The contributions of this article are as follows:

- A neural network for stock prediction is designed which can deal with stock ID feature (sparse feature) *via* an embedding structure and dense features.
- We propose a hybrid model which ensembles the output of neural network and Tabnet and it outperforms other models like Lightgbm, Xgboost and separate models.
- Our work is publicly available for other researchers for further research.

## ALGORITHM PRINCIPLE AND STRUCTURE

This chapter introduces the principles, structures, advantages, and disadvantages of the relevant algorithms utilized in this article.

### Xgboost algorithm

Based on the boosting framework, the XGBoost algorithm can be regarded as an improved version of the GBDT model. This algorithm takes the decision tree as the base function and combines the additive model (the combination of weak learners) with the forward distribution algorithm to realize the optimization process of learning. Specifically, the basic idea of the XGBoost algorithm is to continuously split the input data to grow a CART tree, where each tree corresponds to a new function to fit the residuals of the previous tree predictions, namely the gradient boosting method (*Friedman, 2001*). Ultimately, the characteristics of the sample correspond to a leaf node in each tree with a predicted value. The predicted result for the sample is obtained by summing these values.

It can be seen from the above that making the value of the objective function (loss function) as close to zero as possible is the core objective of the XGBoost algorithm. The XGBoost algorithm improves upon the loss function from the gradient boosting algorithm, becoming a more efficient boosting algorithm. The loss function (*Chen & Guestrin, 2016*) can be written as:

$$Obj = \sum_{i=1}^{m} l\left(y_i', y_i\right) + \sum_{k=1}^{k} \Omega\left(f_k\right), \tag{1}$$

where $i$ represents the ith sample in the data set; $m$ is the total amount of data in the kth tree; $k$ stands for the kth CART tree generated in the model. $y_i'$ and $y_i$ represents the predicted value and the real value respectively. $l$ is the empirical loss function of based-tree model, which is mainly divided into square loss function and logistic regression loss function.

Different from GBDT algorithm, the regular term $\Omega$ is added to the loss function of XGBoost model to control the complexity of the model and reduce the possibility of over fitting, where $\Omega$ can be written as:

$$\Omega = \gamma T + \frac{1}{2}\lambda \|w\|^2, \tag{2}$$

where $T$ represents the number of nodes in the CART tree, $\frac{1}{2}\|w\|^2$ means L2 regularization of the value $w$ of the node. $\gamma$ and $\lambda$ are super parameters, which are used to control the weight of $T$ and $w$. The mapping relationship q between the weight vector $w$ and the leaf node is as follows:

$$f_t(\mathbf{x}) = w_{q(x)}, w \in \mathbf{R}^T, q : \mathbf{R}^D \to \{1, 2, \ldots, T\}, \tag{3}$$

Because XGBoost model is essentially an additive model, its final prediction score is the cumulative sum of the scores of each weak learner. Assuming that the CART tree generated in iteration $t$ is ft, there is:

$$y_i^{'(t)} = \sum_{k=1}^{t} f_k(\mathbf{x}_i) = y_i^{'(t-1)} + f_t(\mathbf{x}_i), \tag{4}$$

Bring Eq. (4) into Eq. (1), and the loss function can be expressed as:

$$Obj^{(t)} = \sum_{i=1}^{n} l\left(y_i, y_i^{'(t)}\right) + \sum_{k=1}^{k} \Omega(f_k) = \sum_{i=1}^{n} l\left[y_i, y_i^{'(t-1)} + f_t(\mathbf{x}_i)\right] + \Omega(f_t), \tag{5}$$

Then, the loss function is approximated to obtain the optimal solution. The traditional GBDT algorithm only uses the first derivative information of the objective function, resulting in greater information loss. The improved XGBoost algorithm performs a Taylor expansion on the objective function, while retaining both the first-order and second-order derivative information. This approach is conducive to more accurate and faster gradient descent. Therefore, the approximate loss function can be expressed as follows:

$$Obj^{(t)} \approx \sum_{i=1}^{n} \left[g_i f_t(\mathbf{x}_i) + \frac{1}{2} h_i f_t^2(\mathbf{x}_i)\right] + \Omega(f_t), \tag{6}$$

where $g_i$ and $h_i$ are the first-order and second-order partial derivatives of the loss function l, respectively. Divide all samples xi belonging to the jth leaf node into a leaf node sample set, that is

$$I_i = \{i | q(\mathbf{x}_i) = j\}, \tag{7}$$

Then combine Eq. (2), Eq. (3) and Eq. (6), we can get:

$$Obj^{(t)} \approx \sum_{i=1}^{n} \left[g_i f_t(x_i) + \frac{1}{2} h_i f_t^2(x_i)\right] + \frac{1}{2}\lambda |w|^2$$

$$= \sum_{j=1}^{T} \left[\left(\sum_{i \in I_i} g_i\right) w_j + \frac{1}{2}\left(\sum_{i \in I_i} h_i + \lambda\right) w_j^2\right] + \gamma T, \tag{8}$$

It can be seen that there is only one variable in Eq. (8), namely, the weight vector $w$. This is because the objective function of each leaf node is completely independent. That is, when the sub-objective functions of each node obtain their optimal values, the final objective function, Obj, can also achieve its optimal value. Once the structure of the model has been determined, the optimal target values of Obj and the leaf node weights $w_j$ can be obtained by deriving Eq. (8), as shown in Eqs. (9) and (10): $w_j$

$$w_j^* = -\frac{G_j}{H_j + \lambda}, \tag{9}$$

$$Obj^{(t)} = -\frac{1}{2}\sum_{j=1}^{T}\frac{G_j^2}{H_j + \lambda} + \gamma T \tag{10}$$

where $G_j$ stands for $\sum_{i \in I_i} g_i$, $H_j$ represents $\sum_{i \in I_i} h_i + \lambda$, both are constants. In addition, the determination of the model structure is consistent with the construction of the decision tree. The greedy algorithm is used to select the sub-CART structure that can minimize the system's entropy; that is, each node segmentation is based on making the increment of the Obj value as large as possible.

Beyond the modifications and optimizations of the GBDT algorithm, the XGBoost algorithm has also undergone significant optimization in engineering aspects, such as supporting parallel operations and effectively using hardware resources. These improvements make it superior to its predecessors in terms of accuracy and operational speed. However, the algorithm still has some shortcomings, such as having too many parameters that are difficult to adjust, being only suitable for dealing with structured data, resulting in a heavy dependency on feature engineering, and an inability to handle high-dimensional feature data, among others.

## LightGBM algorithm

To address the shortcomings of the XGBoost algorithm, the LightGBM algorithm has been proposed. The XGBoost algorithm is optimized by the LightGBM algorithm in the following three aspects (*Ke et al., 2017*): (1) using a histogram-based algorithm to tackle the difficulty of too many split points in the CART tree; (2) adopting gradient-based one-side sampling (GOSS) to address the issue of having too many samples; (3) employing exclusive feature bundling (EFB) to resolve the problem of excessive features.

Through histogram statistics, large-scale data is placed in the histogram. The basic framework is shown in Fig. 1.

Compared with the XGBoost algorithm, which needs 32-bit floating-point numbers to store characteristic values, the histogram algorithm can generally use 8-bit integers to store the discrete values of characteristics. Therefore, the memory usage can be saved. The histogram only requires calculating the gain for $k$ node splits, reducing the time complexity from O(Number of eigenvalues * features) to O($k$ * features), which significantly improves the running speed. A small gradient usually indicates a small training error. Training the

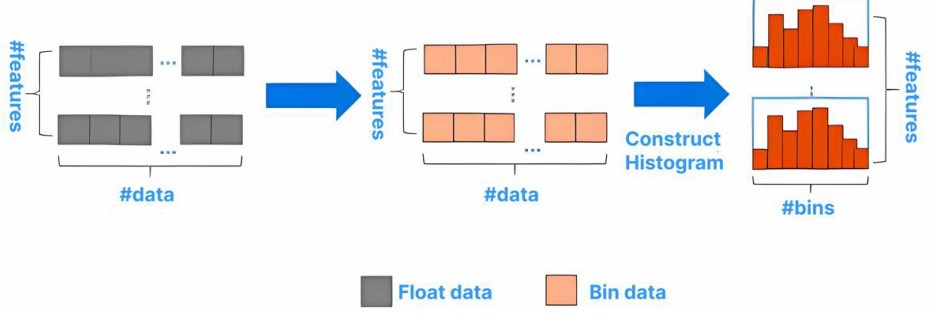

**Figure 1** Basic flow of histogram algorithm.

model with data having small gradients does not improve the model's accuracy but can reduce efficiency with only a slight increase in running time. However, directly discarding this portion of data with small gradients would alter the data distribution and reduce the model's accuracy. The GOSS algorithm effectively addresses this issue. It samples according to the weight information, ensuring that the data distribution is not significantly changed while reducing a large number of samples with small gradients. Specifically, the algorithm selects a% of the data with larger gradients, then randomly selects b% of the data with smaller gradients, and multiplies the b% data by a coefficient (1-a)/b. This approach ensures the algorithm pays more attention to samples with small gradients without significantly altering the original dataset's distribution.

The EFB algorithm is a lossless method for reducing feature dimensions. It operates by calculating the conflict ratio between different features to measure the degree of non-exclusivity. A low conflict ratio indicates a high level of mutual exclusivity. Features with high mutual exclusivity can be bundled and fused, thus reducing the number of features and improving operational efficiency.

## DNN model

The essence of DNN is to build a network structure with multiple hidden layers based on massive data (see Fig. 2) to ultimately enhance model performance. In contrast to manual rule selection or the construction of important features, DNN can retain more information from the data and reduce dependence on feature engineering.

The two adjacent layers of neurons are fully connected, and the initial weights are generated through unsupervised pre-training. The output results are applied as the input data for the next hidden layer and are carried out successively until the final results are output; this process is known as forward propagation. In addition, the weight value of each hidden layer is obtained by error back-propagation.

In the forward propagation algorithm, the mapping relationship between the two connected layers can be written as follows:

$$z = \sum_{i=1}^{m} w_i x_i + b \tag{11}$$

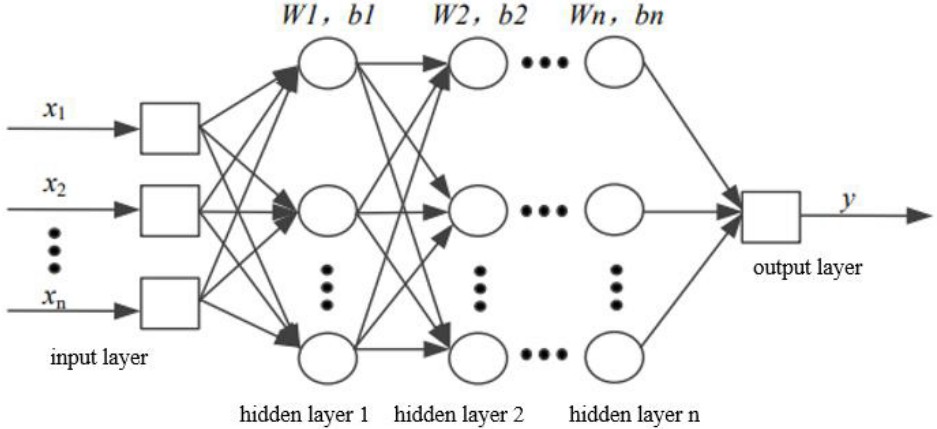

**Figure 2   Structural diagram of DNN.**

where m means the number of input neurons, $x_i$ represents the output data of the previous neuron, w represents the vector of weight, and b stands for the offset term. In order to prevent the output from increasing infinitely, an activation function $\sigma(z)$ is added, such as tanx, softmax, ReLU and so on. At the same time, Eq. (11) is expanded. Assuming that there are m neurons in layer $l-1$, the output of the jth neuron in layer $l$ can be expressed as:

$$a_j^l = \sigma\left(z_j^l\right) = \sigma(\sum_{k=1}^{m} w_{jk}^l a_k^{l-1} + b_j^l) \tag{12}$$

Assuming that there are n neurons in layer $l$, by extending Eq. (12) and using matrix method, the output of layer $l$ can be obtained:

$$\sigma^l = \sigma\left(z^l\right) = \sigma(W^l a^{l-1} + b^l) \tag{13}$$

where $W^l$ represents the weight coefficient matrix with the size of n × m in the $l$ layer, and $b^l$ represents the offset vector with the size of n × 1.

The error back-propagation algorithm involves selecting a loss function to measure the output loss of samples during model training and minimizing this loss function. The corresponding results are the coefficient matrix

W and the offset vector b. In DNNs, the process of finding the optimal values is generally completed through iterations of the gradient descent method.

## TabNet model

To enhance predictive power, traditional DNNs often blindly increase the number of network layers, leading to the over-fitting of the model. The TabNet model is a neural network structure with a decision manifold similar to that of a tree model. It has realized the process of calculating the gain of each feature separately when the tree splits, as shown in Fig. 3 (*Arik & Pfister, 2021*).

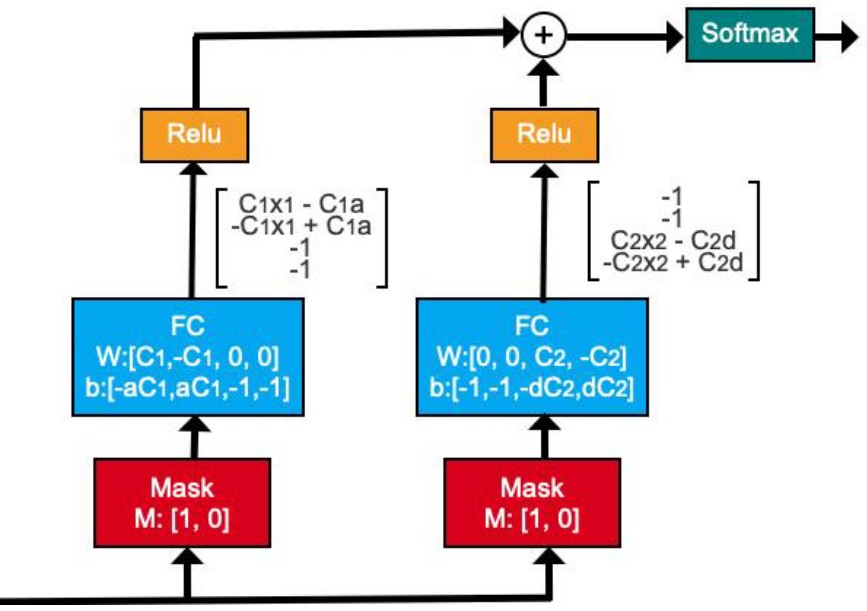

**Figure 3** **Illustration of decision manifold using DNN structures.**

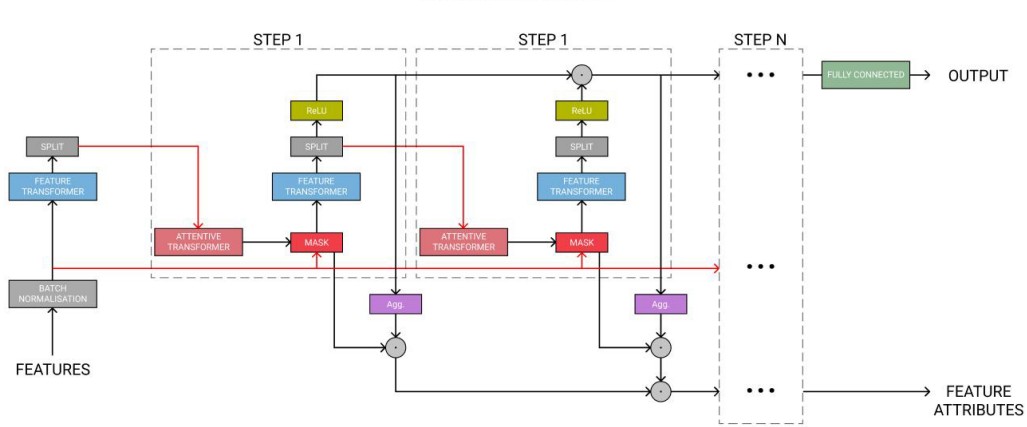

**Figure 4** **TabNet encoder architecture.**

Next, Fig. 4 displays the complete TabNet encoder architecture. The original data are normalized through the Batch Normalization module and then processed in multiple sequential steps, such as STEP1, STEP2, *etc*.

The feature transformer module functions to extract features, enabling the extraction of more effective information representations for sample attributes. Then, the obtained information is divided into two parts by the SPLIT module. One part is used for the current output, and the other serves as the input for the next step.

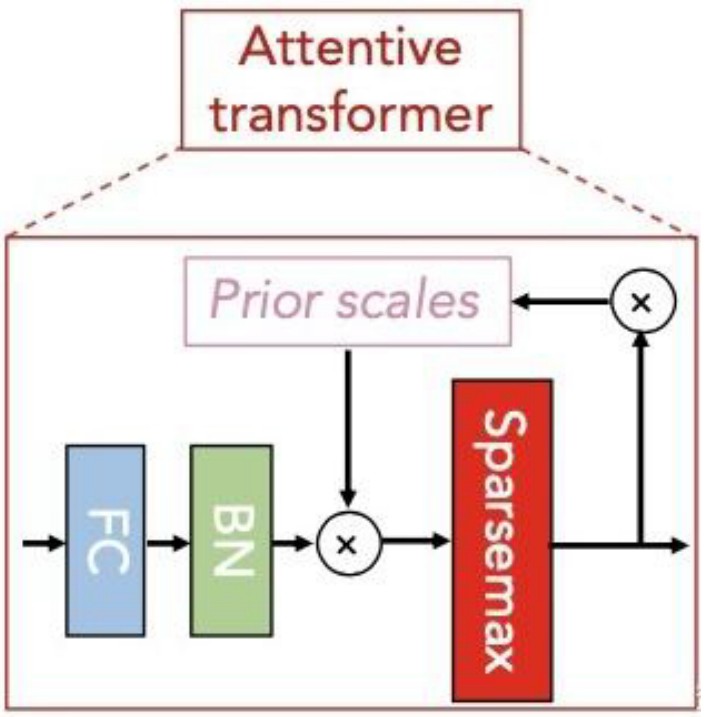

**Figure 5  Attentive transformer module.**

Subsequently, the attentive transformer module learns the importance of each feature in each sample and obtains the corresponding mask matrix to filter out the unimportant features. Figure 5 shows the basic structure of this module. The formula based on Fig. 5 is expressed as follows:

$$M[i] = \text{Sparsemax}(P[i-1] \cdot h_i(a[i-1])) \tag{14}$$

In Eq. (14), a[i-1] is the characteristic information divided by SPLIT in the last step decision, $h_i$ represents FC and BN layerS, P[i-1] is the priors scales item, Sparsemax is similar to softmax, which can obtain more sparse output results. Where, $P[i] = \prod_{j=1}^{i}(\gamma - M[j])$, is used to indicate the degree of use of features in the previous steps. When $\gamma$ is smaller, the features selection is sparser.

According to the properties of Sparsemax, we can obtain:

$$\sum_{j=1}^{D} M[i]_{b,j} = 1 \tag{15}$$

Therefore, M[i] can be understood as the attention weight distribution of the $D$-dimensional feature for each input sample at the current step of the model. For different samples, the output attention weight varies, a characteristic referred to as instance-wise. Based on this characteristic, the TabNet algorithm can select different features for different samples, an improvement over the tree-based model, which does not have this ability.

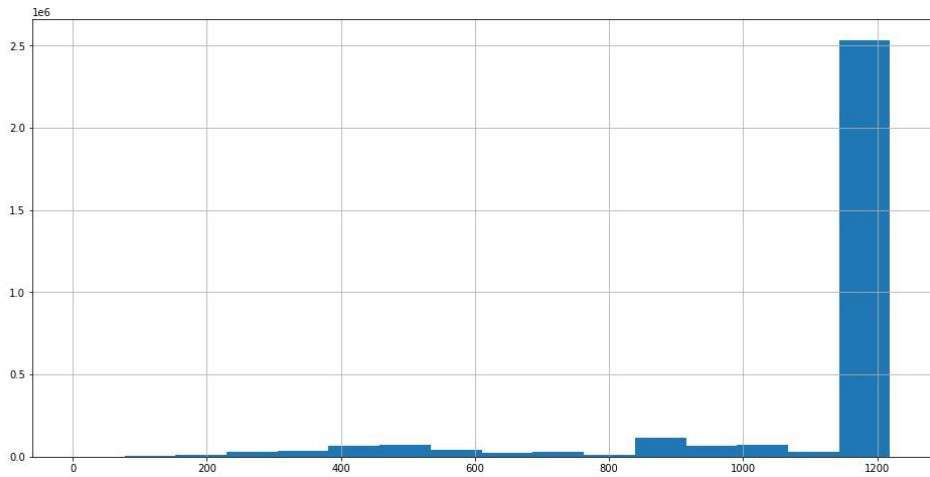

**Figure 6  Time span distribution of different stock data.**

The TabNet algorithm inherits the advantages of DNN, including representation learning and end-to-end training, which greatly reduces the dependence on feature engineering. At the same time, it combines the advantages of tree-based methods, including strong interpretability and sparse feature selection, thereby improving accuracy and efficiency.

# DATA AND PROPOSED MODEL

## Data set introduction and analysis

The data adopted of this article contains the features of real historical data from thousands of investments, and the main fields include time_id, investment_id (3,587 different stocks), target (return on investment) and 300 anonymous features generated from market data. The investment_ id belongs to ID feature, and the others belong to numerical features. Before model training, these two types of features need to be processed with different encoding methods. Embedding algorithm has the ability to decrease the dimension of ID feature, while ensuring the integrity of its information; due to the low dimension of other numerical features, One-hot encoding can be adopted.

In addition, due to commercial confidentiality, data exists in the form of processing and encryption.

Through the statistical analysis of different stocks in the data, we can see that the time span of most stock data is about 1,200 days from Fig. 6, which is conducive to improving the prediction accuracy of the model with uniform distribution data.

Figure 7 shows the distribution of stock returns. It can be seen that the data are mainly concentrated in the area with small absolute values of returns, and the returns away from the $x$-axis occupy only a small amount of data. The distribution of features is consistent with the distribution of stock returns, as shown in Fig. 8, which indicates that the fifth feature has a positive correlation with stock returns.
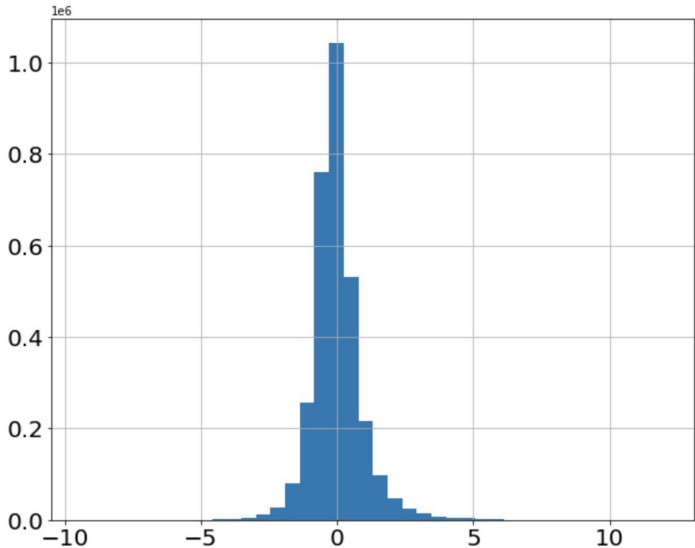

**Figure 7   Distribution of stock returns.**

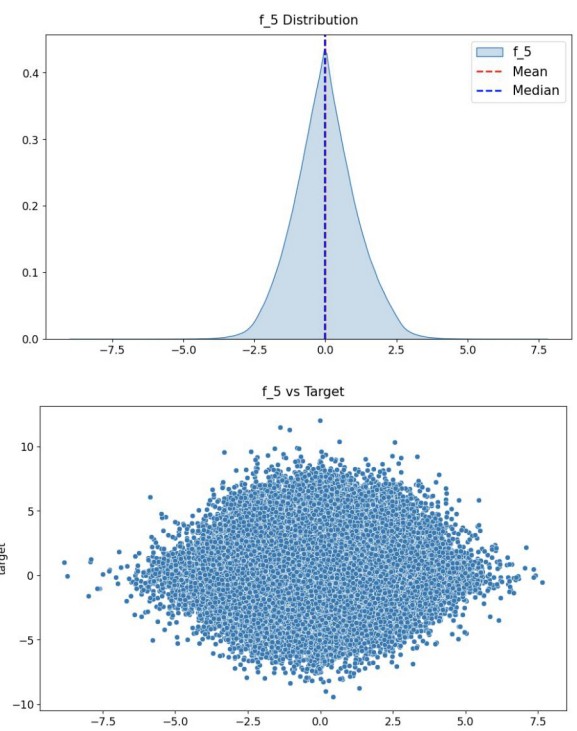

**Figure 8   The distribution of the fifth feature (top) and the relationship between this feature and the prediction target (bottom).**

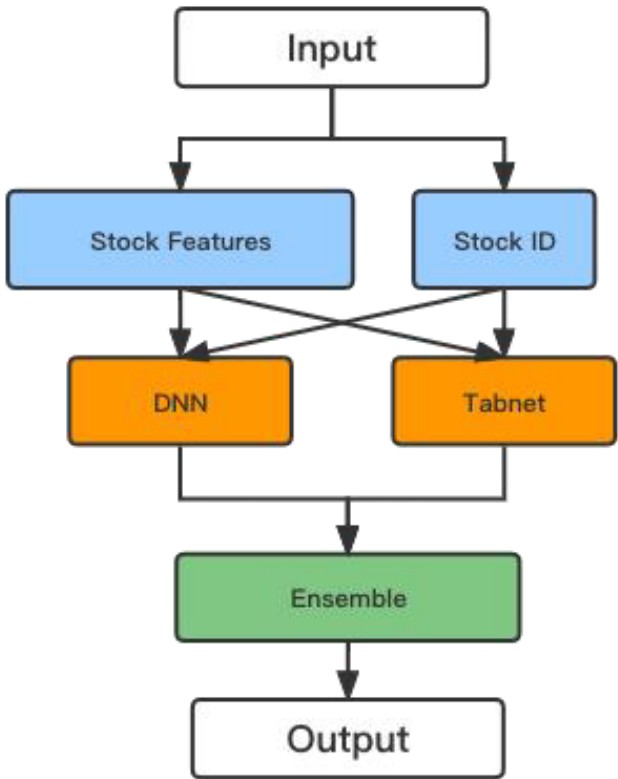

**Figure 9** **The flow chart of the hybrid model.**

### Proposed model

(1) Model building

This article proposes a hybrid model based on the DNN and TabNet algorithms to calculate stock returns and prices. Figure 9 shows the flowchart of the hybrid model.

After encoding the ID and numerical features in the initial data so that their dimensions do not differ too much, the data are input into the DNN and TabNet models for training, respectively. Finally, the predictions from the two models are combined to yield more accurate values.

The DNN model is constructed using fully connected layers with the Swish activation function. The DNN structure used in this article is shown in Fig. 10.

After encoding and dimensionality reduction of two different types of data, we extract and learn the associations between features through three fully connected layers (dense layers). We combine the features learned above and then output the final result through three Dense layers. In addition, to enhance the robustness of the DNN model and reduce the probability of overfitting, Gaussian noise is added to the input numerical data, which has little impact on the predicted results.

The structure of the TabNet model has been introduced in detail in 'Algorithm Principle and Structure'.

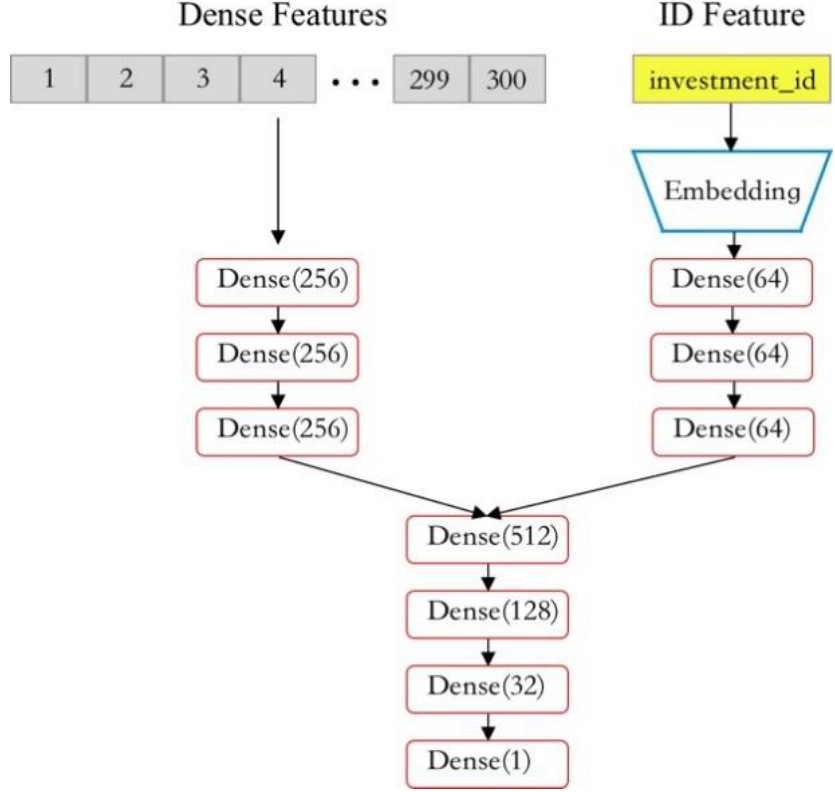

**Figure 10  Structure of DNN model.**

Like other neural network models, the TabNet algorithm is very sensitive to hyperparameters, so adjusting and selecting appropriate parameters is important. Generally speaking, the parameters that greatly impact TabNet include N_steps, feature_dim, gamma, lambda sparsity, *etc*. Among them, larger data and more complex tasks require larger N_steps, but this may lead to overfitting. Adjusting the values of feature_dim and output_dim is the most effective way to balance performance and complexity. Through adjustment and optimization, the selection of model parameters in this article is shown in Table 1.

## EXPERIMENTAL RESULTS AND ANALYSIS

### PCC evaluation index
In order to quantitatively evaluate the performance of our model and others, the Pearson correlation coefficient (PCC) is selected as the evaluation index. This indicator measures the correlation between two variables, x and y, with its value ranging between −1 and 1. Generally speaking, the larger the PCC value, the closer it is to 1, indicating a greater positive correlation between the two variables. The calculation expression can be written as follows:

$$\rho_{X,Y} = \frac{cov(X,Y)}{\sigma_X \sigma_Y} = \frac{E[(X-\mu_X)(Y-\mu_Y)]}{\sigma_X \sigma_Y} \tag{16}$$

**Table 1  Specific setting of parameters.**

| Settings | Values |
|---|---|
| N_steps | 2 |
| feature_dim | 16 |
| output_dim | 16 |
| gamma | 1.4690246 |
| lambda-sparsity | 0 |
| batch_ size | 1024 |
| epoch | 20 |
| learning rate | 0.0001 |
| optimizer | Adam |

**Table 2  PCCs values of different algorithms.**

| Models | PCCs |
|---|---|
| Proposed model | 0.1269 |
| LightGBM | 0.1145 |
| Xgboost | 0.1102 |
| DNN | 0.1103 |
| TabNet | 0.1205 |

where $\sigma_X$ and $\sigma_Y$ are the standard deviations of variables X and Y respectively.

## RMSE evaluation index

Root mean square error (RMSE) index is a common index for regression task as shown in the following formula:

$$\text{RMSE} = \sqrt{\frac{1}{n}(Y - \hat{Y})^T (Y - \hat{Y})} \tag{17}$$

where Y is the vector of ground truth and $\hat{Y}$ is the prediction vector and n is the sample number.

## Result analysis

The PCC between the predicted results of stock returns obtained by different models and the real values is calculated (see Table 2). The larger the PCC value, the closer the predicted result is to the real return. To some extent, this index can express the accuracy of the model's predictions.

Obviously, our hybrid algorithm has the highest PCCs value of 0.1269, which is better than the separate DNN and TabNet models, which are 0.1103, 0.1205 respectively. Then, the tree-based algorithms are compared, with the PCC values of the XGBoost algorithm and the LightGBM algorithm reaching 0.1102 and 0.1145, respectively. The combination of the DNN algorithm and the TabNet model makes the prediction accuracy on tabular data or time series exceed that of tree-based algorithms such as XGBoost and LightGBM, which have traditionally held a dominant position. This indicates that the model we propose performs exceptionally well on large datasets with minimal feature engineering

**Table 3    RMSE values of different algorithms.**

| Models | RMSE |
|---|---|
| Proposed model | 0.8862 |
| LightGBM | 0.9124 |
| Xgboost | 0.9250 |
| DNN | 0.9110 |
| TabNet | 0.9021 |

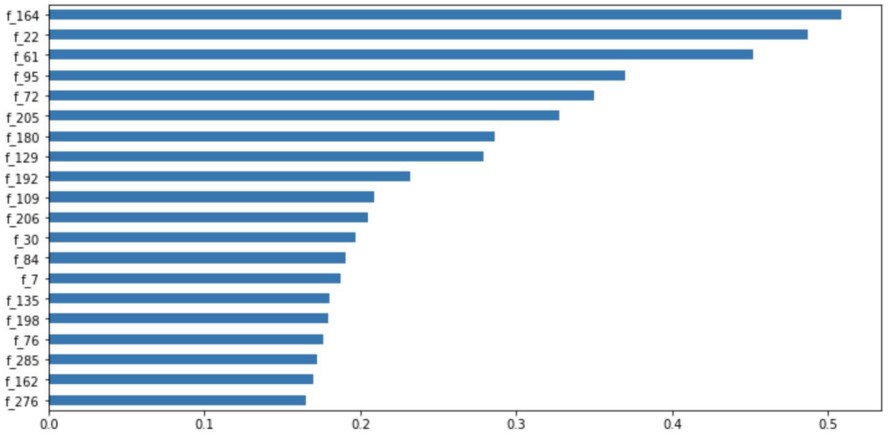

**Figure 11    Feature importance ranking.**

required. The deep learning method has become a powerful tool for enhancing the model's performance in predicting stock prices and returns.

Except to PCCs metric, another metric RMSE for evaluating the accuracy of the proposed method. RMSE is a metric to evaluate the regression task and a smaller RMSE means a better performance. Thus, from Table 3, proposed model achieves a lowest RMSE score, which means beating other single models.

In addition, the TabNet model, based on the attention transformer and instance-wise processing, has a strong explanatory capability and enhanced learning ability. This is attributed to the selection of the most influential and significant features at each decision step. The model can quantify the contribution of each feature to the training model, as shown in Fig. 11. Among the given 300 features, the 164th, 22nd, and 61st features evidently have the greatest impact on the performance of our model. According to the results in Fig. 11, during model training, we can filter out the features that have little or no impact on the performance of the model and enhance the learning of the features that have a greater impact, thereby obtaining a more accurate prediction result.

## CONCLUSIONS

This article primarily constructs a hybrid model for stock price and return prediction based on the DNN and TabNet models to compensate for the deficiencies of neural

network algorithms in processing tabular and time series data. The TabNet algorithm constructs a neural network with a decision manifold similar to that of tree-based models, retaining the end-to-end and representation learning characteristics of deep learning while inheriting the interpretability of tree-based models and the advantages of sparse feature selection. The algorithm is combined with the DNN model to further improve the model's performance. The results show that our algorithm has the highest PCCs value, which is 15.15% and 10.83% higher than the XGBoost and LightGBM algorithms, respectively. The performance of the proposed algorithm surpasses that of the dominant tree-based models. In addition, our algorithm performs particularly well on large datasets with minimal feature engineering and has strong interpretability, such as quantifying the contribution of different features in the model. This has significant research importance and broad application prospects.

Although we have achieved relatively good results in the task of stock prediction, there is still room for improvement. Future work will focus on several aspects: (1) exploring the integration of more models; (2) using different model fusion strategies; (3) conducting feature selection based on feature importance and the similarity between features, to filter out truly useful features.

### Funding

The authors received no funding for this work.

### Competing Interests

The authors declare there are no competing interests.

### Author Contributions

- Tonghui Zhang conceived and designed the experiments, performed the experiments, analyzed the data, performed the computation work, authored or reviewed drafts of the article, and approved the final draft.
- Ming Da Huo conceived and designed the experiments, performed the experiments, analyzed the data, performed the computation work, authored or reviewed drafts of the article, and approved the final draft.
- Zhaozhao Ma analyzed the data, performed the computation work, prepared figures and/or tables, and approved the final draft.
- Jiajun Hu performed the experiments, performed the computation work, authored or reviewed drafts of the article, and approved the final draft.
- Qian Liang analyzed the data, performed the computation work, prepared figures and/or tables, and approved the final draft.
- Heng Chen conceived and designed the experiments, performed the computation work, authored or reviewed drafts of the article, and approved the final draft.

### Data Availability

The data is available at figshare: Bat, Catch (2023). Peej_dataset. figshare. Dataset. https://doi.org/10.6084/m9.figshare.24616410.v2.

## Supplemental Information

Supplemental information for this article can be found online at http://dx.doi.org/10.7717/peerj-cs.2057#supplemental-information.

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
