# Peer review of "Prediction model of stock return on investment based on hybrid DNN and TabNet model"

_PeerJ Computer Science, doi:10.7717/peerj-cs.2057_

## Round 0.1 · original submission · Major Revisions

Please review and address all the comments, especially make sure that the format of the references match the requirements of PeerJ.

**Language Note:** The review process has identified that the English language must be improved. PeerJ can provide language editing services - please contact us at [email protected] for pricing (be sure to provide your manuscript number and title). Alternatively, you should make your own arrangements to improve the language quality and provide details in your response letter. – PeerJ Staff

·

Basic reporting

The structure of the summary is very good, but the structure of the introduction needs to be adjusted and optimized. Background does not highlight the importance of their own research, the relevant literature review has not been very well summarized. Innovation is not particularly well described.
Part 2 The overall structure of the algorithm is ok, but some of the English descriptions are wrong.

Experimental design

The evaluation indicators in Part 4 are too single and it is recommended to use indicators. At the same time, the result analysis is too simple, it is recommended to further deepen and detailed.
The models introduced in Part 3 are suggested for drawing descriptions.

Validity of the findings

The structure of Part 5 needs to be optimized. There is no indication of what further research can be carried out

Reviewer 2 ·

Basic reporting

This manuscript is not ready for review. The phrase “Reference source not found” appears throughout the manuscript. Citations are not in order. For example, Cuckoo Seach (CS) is used in the text after reference [5] but number 28 in the reference list. The ‘ on the variable y’ used for predicted values should not be a superscript. So far as I can determine, figure 4 is the only figure called out in the text. Most of the text is a review of methods. Discussion of results is less than 1 page and contains not figures or table.

Experimental design

Non given.

Validity of the findings

finding not reported in text

·

Basic reporting

The paper presents the analysis the advantage and disadvantage of the various algorithms for stock market prediction. The concept is good but validation and strong comparison with existing work are required. Abstract should be reframe with applied methodology, dataset and accuracy result.

Experimental design

• what is Obj in equ (1).
• The methodology is meshed and not clear. If possible put a block diagram for the step wise methodology.
• In line 348, should specify the name of the other neural network.
• Some abbreviation like DNN, SVM, PCCS are used in abstract part and Title part also. Need to full form first time.

Validity of the findings

• Put proper justification why XGBoost is applied whereas other learning algorithms are available. What is the value for RMSE in XGBoost.
• How the considerable features are co-related to the target attribute. It will be good if put a hitmap to justify this.
• In line no 203, mentioned “XGBoost model is essentially an additive model, its final prediction score is the cumulative sum of the scores of each weak learner”. How it is suitable for the proposed problem.
• Put citation for the Dataset which is used for validation.

Additional comments

• Need more recent research papers. It will be good if add a section for literature survey.
• The authors have used high quality resolution figures for describing the research work. However, some figures must be enlarged for clear understanding.
• The paper should be prepared according to the template. The numbering for the heading and sub heading should have a sequence.

·

Basic reporting

Resolve reference sources throughout document - confirm that data used is available.

Line 47, sentence "In recent years,..." - clarify. The use of the researchers is confusing. Suggest revision of "In recent years, more and more machine learning algorithms have been used to study and predict the stock return and price."

Line 51, sentence "Using SVM algorithm...", suggest revision to "Using SVM algorithm to predict the future trend, trading volume, profit margin, and other indicators of the stock market [2], the prediction accuracy is ahead of the traditional time series methods [3]"

Line 54, sentence "Devi et al. comprehensively considered the impact of different SVM parameters on the model prediction results, and combined with Cuckoo Search (CS) technology to adjust and optimize the model parameters." suggest removal of "with"

Line 61-65, this is great - clearly describes the issues with current models.

Line 69, is GBDT an acronym? If so, define.

Line 86, keep consistent capitalization for LightGBM

Line 155, sentence "In addition..." change good to better

Line 161 correct spelling, 'performance'

Move main research contents to the top?

Line 182, keep capitalization consistent when discussing models.

Line 192, I do not see a 'm' variable in the eqn - clarify.

Line 228, font change?

Line 230, add space between 'the' and 'former'

Line 230, sentence "However, the algorithm still has..." suggest removing first 'and', resulting in '...only suitable for dealing with structured data, more dependent on Feature Engineering...'

Line 235, adjust sentence to 'The XGBoost algorithm is optimized by the LightGBM algorithm in the following three aspects:...'

Line 243, is the memory usage 1/8, or is it 1/4?

Line 253, should be 'would' instead of 'wilouldl'?

Line 285, change 'which is easy to lead to over fitting of the model.' to 'leading to the over-fitting of the model.'

Line 288 - didn't see Figure 4 in text?

Line 307, change sentence 'TabNet algorithm can select different features for different samples, with the tree-based model not have this ability.' to 'TabNet algorithm can select different features for different samples, an improvement over the tree-based model, which does not have this ability.'

Line 322, remove 'Besides'

Line 343, adjust '...for enhancing...' to '...to enhance...

Line 346, add 'will', such as ''...and will not be repeated...'

Line 350, '...a great impact on the TabNet include...' change to ''...a great impact on TabNet, including..."

Line 367, remove one of the double periods

In line 370, suggest listing the separate DNN and TabNet PCC values

Line 392, manifolds, not manifoldS?

Line 395, add these percentages to the results paragraph.

Line 401, font change?

In conclusion, may want to mention model limitations/challenges, that can lead into future work.

Experimental design

No comment

Validity of the findings

No comment

---

## Round 0.2 · Major Revisions

The latest iteration exhibits a notable enhancement in quality compared to its predecessor. Nonetheless, certain misrepresentations, such as "our hybrid algorithm has the highest PCCs value of 0.1269, which is better than the separate DNN and TabNet models, which are 0.1492, 0.1472 respectively," may compromise the credibility of your findings. It is imperative to thoroughly review the entire article and all accompanying documents to ensure the accuracy of the figures presented.

·

Basic reporting

No subheadings are needed in the introduction, but the structure needs to be clear, and there must be logic and distinctions between paragraphs.
Two evaluation indicators are used but not introduced in the RMSE.

Experimental design

Robustness tests need to be done

Validity of the findings

The paper is modified with major improvements.

Reviewer 2 ·

Basic reporting

meets all these

Experimental design

meets all these requirements

Validity of the findings

meets all these requirements

Additional comments

line 384: our hybrid algorithm has the highest PCCs value of 0.1269, which is better than the separate DNN and TabNet models, which are 0.1492, 0.1472 respectively. inconsistent statement needs correction.

·

Basic reporting

Line 51: double period after "...industry", resolve

Line 73: Switch acronym definition format to "Gradient Boosting Decision Tree (GBDT)" for consistency with the RF definition

Line 77: Use RF for random forest, as previously defined

Line 78: Should linear regression be capitalized? If so, use the same capitalization when mentioned in Line 77

Line 80: Use RF for random forest, as previously defined

Line 83: Use GBDT only, as previously defined

Line 99: Use RF for random forest, as previously defined

Line 109: May now use BPNN, as this acronym has been defined

Line 116: May now use BPNN, as this acronym has been defined

Line 118: Use BPNN instead of "BP neural networks"

Line 123: Use WNN, as previously defined

Line 125: Use CS for Cuckoo Search, as previously defined

Line 128: Use DNN, as previously defined - or remove from Line 106 and allow this to be the first introduction

Line 145 to 146: Use DNN, CNN, and RNN, as previously defined

Line 154: Recommend re-organization to "Long Short-Term Memory (LSTM)" for consistency

Line 173: GRU previously defined?

Line 199: Is CART an acronym - if so, define

Line 214: Capitalize CART, for consistency

Line 230: Fix formatting of the weight vector

Line 253: Lowercase "Employing"

Line 348: double period after "...returns", resolve

Line 374: Capitalize "Pearson Correlation Coefficient", consistent with the acronym defined

Line 381: Use PCC, as defined

Line 393: Recommend re-organization to "Root Mean Square Error (RMSE)" for consistency

Line 412: Use "PCC" only, as acronym previously defined

Experimental design

No comment

Validity of the findings

Line 260 to 261: Can increase efficiency, not reduce? (if training the model with small gradients doesn't improve accuracy or efficiency (increased run time) - why do it? Reduces error?) - clarify

Line 269: EFB is already defined as "Exclusive Feature Building" (Line 252), not "Enhanced Feature Building" - resolve

Line 384 to 385: Hybrid algorithm PCC value is lower than DNN and TabNet PCC values listed. This is inconsistent with the text - resolve

Line 413: double check these percentages...

---

## Round 0.3 · accepted · Accept

I confirm that the authors have effectively addressed all of the reviewers' comments. Based on the current version, I believe this manuscript is ready for publication.